# CD137^+^ T-Cells: Protagonists of the Immunotherapy Revolution

**DOI:** 10.3390/cancers13030456

**Published:** 2021-01-26

**Authors:** Alessio Ugolini, Marianna Nuti

**Affiliations:** 1Department of Experimental Medicine, “Sapienza” University of Rome, Viale Regina Elena, 324-00161 Rome, Italy; alessio.ugolini@moffitt.org; 2Department of Immunology, H. Lee Moffitt Cancer Center, Tampa, FL 33612, USA

**Keywords:** CD137, 4-1BB, TILs, CD137^+^ T-cells, immunotherapy, ACT, CAR-T, monoclonal antibodies, biomarker

## Abstract

**Simple Summary:**

The CD137 receptor is expressed by activated antigen-specific T-cells. CD137^+^ T-cells were identified inside TILs and PBMCs of different tumor types and have proven to be the naturally occurring antitumor effector cells, capable of expressing a wide variability in terms of TCR specificity against both shared and neoantigenic tumor-derived peptides. The aim of this review is thus summarizing and highlighting their role as drivers of patients’ immune responses in anticancer therapies as well as their potential role in future and current strategies of immunotherapy.

**Abstract:**

The CD137 receptor (4-1BB, TNF RSF9) is an activation induced molecule expressed by antigen-specific T-cells. The engagement with its ligand, CD137L, is capable of increasing T-cell survival, proliferation, and cytokine production. This allowed to identify the CD137^+^ T-cells as the real tumor-specific activated T-cell population. In fact, these cells express various TCRs that are specific for a wide range of tumor-derived peptides, both shared and neoantigenic ones. Moreover, their prevalence in sites close to the tumor and their unicity in killing cancer cells both in vitro and in vivo, raised particular interest in studying their potential role in different strategies of immunotherapy. They indeed showed to be a reliable marker able to predict patient’s outcome to immune-based therapies as well as monitor their response. In addition, the possibility of isolating and expanding this population, turned promising in order to generate effector antitumor T-cells in the context of adoptive T-cell therapies. CD137-targeting monoclonal antibodies have already shown their antitumor efficacy in cancer patients and a number of clinical trials are thus ongoing to test their possible introduction in different combination approaches of immunotherapy. Finally, the intracellular domain of the CD137 receptor was introduced in the anti-CD19 CAR-T cells that were approved by FDA for the treatment of pediatric B-cell leukemia and refractory B-cell lymphoma.

## 1. Introduction

Immunotherapy aims to re-educate the patient’s immune system to recognize and fight cancer cells. The existence of T-cells with a potential antitumor effect has laid the foundation for most of the current approaches of immunotherapy. In fact, the use of therapies such as immune checkpoint inhibitors (ICIs), DC vaccines, and adoptive T-cell transfer (ACT) finally relies on the presence of a population of effector T-cells that is capable of killing tumor cells. These immune-based drugs thus aim to unleash this population from different regulatory constraints such as T-cell exhaustion or the impossibility of reaching cancer cells, to subsequently limit tumor growth and progression. As a confirmation, the accumulation of tumor-infiltrating lymphocytes (TILs) correlates with a better clinical outcome and an improved survival in most tumor models [1,2,3,4,5,6,7,8,9,10,11], indicating their importance in predicting patients response to anticancer therapies. Nevertheless, the composition of TILs is heterogeneous [12] and it still remains challenging to identify the real population of naturally occurring antitumor T-cells [13]. Therefore, this review will discuss the emerging role of the CD137^+^ T-cells population as the main effector population activated against cancer cells with all the possible implications for the future of immunotherapy.

## 2. CD137: The Receptor

The CD137 receptor (4-1BB, TNFRSF9) is a member of the tumor necrosis factor receptors (TNFR) family and was characterized as an inducible costimulatory receptor on T-cells, together with its ligand (CD137L, 4-1BBL), both in human and mice [14]. CD137 was initially described as a surface marker expressed by activated T-cells, with an in vitro peak expression 48 h after the primary T-cell activation signal and a decline starting from day 4–5 [14,15,16,17]. In vivo, its expression upon activation turned out to happen even earlier, starting indeed at 12h post-immunization [18,19]. Both CD4^+^ and CD8^+^ T-cells are able to upregulate CD137, even if its expression on CD8^+^ T-cells is earlier and higher [20,21,22]. 

However, CD137 receptor is not a specific marker for T-cells, since it can be expressed, even if to a smaller extent, also by dendritic cells (DCs), monocytes, natural killer (NK) cells, eosinophils, and microglia [23]. On the other side, CD137L is expressed by activated antigen presenting cells (APC) as macrophages, DCs, and B-cells [20,23,24,25,26]. Therefore, it is reliable to suppose that the engagement between CD137 and its ligand is part of the complex pathways of interactions between APCs and T-cells.

Similarly to other members of the TNFR family, the CD137 receptor relies on TRAFs proteins to build its signaling [27]. The binding of both CD137L [28,29] and agonistic antibodies [30] results in a quick recruitment of TRAF1 and TRAF2 to the receptor. The consequent TRAF-mediated activation of NF-kB and MAPK intracellular signaling, leads to T-cell division and proliferation, an increased cell survival and enhanced effector functions in both CD4^+^ and CD8^+^ T-cells [15]. As for CD137 expression, also TRAF1 expression is induced by T-cell activation, confirming that the CD137-induced signalosome is required for cytotoxic T-cells (CTL) expansion and for the boosting of effector functions [27]. In fact, a number of mice experiments proved that CD137 stimulation is able to increase T-cell proliferation and cytokine production [14,23]. Consistently, in absence of the CD28 signal, T-cells treated with an anti-CD3 and CD137L can proliferate and produce interleukin 2 (IL-2) to a similar extent of those treated with the combination of anti-CD3 and anti-CD28, but just in the presence of a consistent antigen stimulation [31,32,33,34]. This evidence was one of the starting points to get to the notion that CD137 identifies those T-cells that are activated against a specific antigen. The CD137L stimulation of human CD8^+^ T-cells leads to the expansion of this T-cells subset which is followed by an increase of effector molecules such as granzyme A, interferon ɣ (IFN-ɣ), perforin, and different cytokines, driving CD8^+^ memory T-cells toward a differentiated effector phenotype [35,36]. In addition, the CD137 receptor seems to have a strong and prevalent role in increasing T-cell survival by preventing activation-induced T-cell death [37,38] and this appears to be in line with the physiological timing of the CD137 signal that is subsequent to the TCR and CD28 mediated signals. The BIM downregulation and the induction of Bcl-XL and Bfl-1 were pointed out as responsible for the inhibition of the activation-induced cell death, after the CD137 engagement [39]. Further studies also showed that the CD137 engagement is able to stimulate the mitochondrial metabolism in order to increase T-cell respiratory capacities [40,41] and to induce DNA demethylation in CD8^+^ T-cells main genes and chromatin reprogramming [42]. As above mentioned, different studies highlighted a preferential role of CD137 in CD8^+^ T cells rather than in CD4^+^ T cells, even if it can be induced on both the T-cell subsets, including CD4^+^ regulatory T-cells (Tregs) [43]. However, the effective function of CD137 signaling on CD4^+^ T-cells is still unclear and may thus not be as physiologically relevant as for CD8^+^ T cells [44].As confirmation, mice deficient for CD137 show an impaired antiviral response mediated by CD8^+^ T cells [45,46,47]. Similarly, mice deficient for CD137L that were adoptively transferred with OT-1 derived CD8^+^ T-cells, showed a marked reduction of these OVA-specific T-cells in both the late primary response and the secondary expansion to OVA/LPS [19]. Moreover, when anti-CD137 monoclonal antibodies or CD137L injections were tested in cancer therapy, a significant benefit in terms of antitumor response was noticed [21,24,38,48,49,50,51,52,53,54,55,56,57]. When CD137 was targeted by in vivo treatments with an agonistic anti-CD137 monoclonal antibody, it resulted in an increased effect on CD8^+^ T-cells expansion and activation with just a modest effect on CD4^+^ T-cells, thus confirming that in vivo its signal affects predominantly this subset of T-cells [50,58].

Finally, as evidence about the importance of the CD137 receptor in marking those T-cells that were activated to eliminate a non-self-antigen, it was reported that the CD137 resulted specifically upregulated after an alloantigen stimulation and, upon CD137 depletion, it was possible to remove alloreactive T-cells during hematopoietic transplantation [59].

An important step forward in the field of tumor immunology, was made when CD137^+^ T-cells were clearly identified as those cells that were terminally differentiated and antigen-specific effector cells, regardless of the antigen specificity [60]. This allowed the isolation of those cells that were considered the real effector cells activated against tumor antigens [61].

## 3. CD137^+^ T-Cells: The Natural Tumor-Specific Population 

The discovery that CD137 is expressed by most of activated and antigen-specific (both against viral and tumor antigens) CD8^+^ T-cells, allowed the isolation of tumor-specific effector T-cells from blood, without knowing the immunogenic epitopes or the MHC-restriction complex. These cells, even if present at low frequencies, were able to kill antigen-expressing cancer cells upon expansion, although this required an ex vivo restimulation with the defined tumor antigen [60,61]. 

This evidence raised a strong interest in investigating this cell repertoire also inside the tumor. In fact, the tumor microenvironment (TME) is enriched for T-cells specific for defined antigens with cytolytic ability against cancer cells [62]. In addition, even if defined antigens are known for different tumor models, exomic sequencing data in different solid tumors proved that cancer cells express a various and heterogeneous set of mutated neo-antigens that are characteristic for every single patient and thus can be recognized by TILs that are able to exert an antitumor response [63]. As confirmation, T-cell receptors (TCRs) isolated from CD137^+^ TILs, showed a reactivity against various mutations of tumor-derived antigens [64]. Given this evidence, the possibility of identifying a tumor-specific T effector population inside the TME without the knowledge of the antigen epitopes seemed very promising.

Initial evidence proved that CD137 is strongly expressed by TILs if compared to spleen- or lymph nodes-derived T-cells and its expression is induced by hypoxia through hypoxia-inducible factor 1α [65].

Recently, Ye et al. decided to investigate the CD137^+^ T-cells population in ovarian cancer patients, comparing three different locations in which this subset of cells could be found: TME, ascites, and peripheral blood [66]. They demonstrated that CD137^+^ T-cells are present in small percentages in the peripheral blood and, to a larger extent, in ascites and even more inside the tumor, showing a progressive hierarchy with the T-cells in a closer proximity to cancer cells expressing the higher percentages of CD137 and then decreasing gradually toward the periphery. Overnight incubation with autologous cancer cells largely increased the percentage of CD137^+^ T-cells and their ability of producing a consistent amount of IFN-γ. Additionally, CD137 expression was further increased when T-cell lines with a known antigen specificity were used. Most importantly, when human TILs and tumor cells where transferred into immunodeficient mice, only CD137^+^ T-cells (but not CD137^−^ T-cells) were able to inhibit tumor growth [66]. Thus, they demonstrated that CD137^+^ T-cells are those cells that naturally show the real antitumor reactivity, confirming also that they represent a subset of newly recruited antitumor T-effector cells, being CD137 expression a rapid and transient event upon specific activation. Overall, this study proposed a novel method to isolate and expand tumor reactive TILs that can be used for adoptive T-cell transfer approaches; the vast heterogeneity of TCRs is indeed conserved with this strategy thus helping to prevent the escape of those tumor cells that do not express a determined antigen or those that express mutated antigens.

These findings suggested the potential role of CD137^+^ T-cells as key contributors of the antitumor immune responses and thus as potential determiners of the success of immunotherapies as well as novel protagonists of immune-based approaches (Figure 1).

Despite these clear results showing the importance of the CD137^+^ T-cell population in eliciting an antitumor response, evidence about the role of these T-cells in oncologic patients have only recently emerged (Table 1).

In 2020, for the first time we provided evidence about the importance of CD137^+^ T-cells in determining the outcome of metastatic non-small cells lung cancer (NSCLC) patients undergoing immunotherapies [67,68]. Patients that were positive for the autoantibody IgM-Rheumatoid Factor (IgM-RF) showed indeed a reduced frequency of CD137^+^ T-cells in peripheral blood and an increased tendency to develop an early progression, in addition to a markedly reduced progression-free survival (PFS) and overall survival (OS) after the anti-PD-1 treatment [68]. In addition, to confirm the importance of this population as an independent prognostic factor, it was reported how a higher percentage of CD137^+^ T-cells in peripheral blood mononuclear cells (PBMC) at baseline, was alone associated with a prolonged OS as well as PFS of patients in treatment with an anti-PD-1 ICI [68].

In addition, in 2018 it was proven that, in metastatic renal clear cell carcinoma (mRCCC) patients undergoing the anti-PD-1 treatment, the percentage of CD137^+^ T-cells decreased during tumor progression [69]. Moreover, patients pretreated with Tyrosin-kinase inhibitor Pazopanib, showed a robust increase in DC activation profile and a subsequent increase of the frequency of CD137^+^ T-cells when compared to Sunitinib [69]. Still in mRCCC, Zizzari et al. demonstrated that CD137^+^ T-cells were positively associated with patients response to TKI [70]. In fact, responder patients showed a markedly higher percentage of this T-cell subset when compared to non-responders. These results highlight the importance of this T-cell subset in oncologic patients response to therapies that require, even if in an indirect way, the immune system’s ability of killing tumor cells. In this scenario, the percentage of this population in peripheral blood (and most likely also in other districts as draining lymph nodes and TME) could serve as a possible biomarker able to identify those patients that would benefit the most from a determinate treatment that relies on T-cells as final effectors. 

Finally, in 2020, indirect evidence of the CD137^+^ T-cells power in determining a prolonged survival for cancer patients came from a study on melanoma patients where it was shown that TNFRSF9 low methylation levels and the subsequent increased mRNA expression at the tumor site, that was prevalently identified inside T-cells, correlated with a better OS of patients as well as a better PFS and response to the anti-PD-1 treatment [71]. TNFRSF9 mRNA expression positively correlated also with the frequency of effector and memory tumor infiltrating lymphocytes, while it was inversely correlated with the frequency of naïve tumor infiltrating lymphocytes [71]. As a confirmation of its power as biomarker for the identification of activated effector T-cells, TNFRSF9 mRNA expression levels positively correlated with an increased IFN-γ signature [71].

These results indicate the potential role of this population as the driver of a successful immunotherapy, thus suggesting the possibility of investigating its presence in patients before undergoing immune-based treatments. In fact, a reduction in its frequency could account for the impossibility of getting a complete or even partial response at least in part of the oncologic patients. In this scenario, strategies aimed at increasing their numbers could be considered at an initial stage, in order to make the patient more prone to efficiently receive an immunotherapeutic treatment.

## 4. CD137^+^ T-Cell-Based Therapies as Possible Protagonists for Future Immunotherapies

As previously mentioned, TILs represent the ideal candidates for adoptive cell transfer immunotherapies since they are enriched for T-cells specific for tumor-derived antigens, showing a vast antigen specificity. Moreover, upon ex vivo restimulation, they show potent antitumor activity and they can be largely expanded in order to get a consistent amount of cells to be administered to the patient [72,73]. Specifically, they raise a particular interest in those tumors that are poorly immunogenic and thus cannot benefit from the usage of immune checkpoint inhibitors as single agents [74]. In fact, it was initially tested in refractory metastatic melanoma patients, in which the use of TILs that were extracted and then rapidly expanded with feeder cells and high doses of IL-2 [75], revealed capable of generating enough cells for reinfusion. It thus resulted also effective in terms of disease control in particular when accompanied by non-myeloablative lymphodepletion and then followed by a treatment with high-dose IL-2 [76]. The efficacy of TILs-based adoptive cell therapy (ACT) was confirmed by different independent trials showing response rates of 40–50% in metastatic melanoma patients, reaching even a complete tumor regression in 10–25% of patients [76,77,78,79,80,81]. It is important to highlight that the results of all these independent studies demonstrate that a consistent part of the responses observed are durable, in particular in those patients showing a complete tumor regression that remained free from disease for many years after the treatment [82,83,84,85].

Additionally, in ovarian cancer, it is possible to isolate and expand a sufficient number of TILs for ACT, with encouraging clinical results in terms of patients outcome [86,87,88,89,90,91]. In addition, a large-scale expansion of TILs was reported for other different solid cancer types, including breast, cervical, colon renal and sarcoma, even if the results in terms of clinical outcome for TIL-based ACT in these subsets of patients were modest [92,93,94,95].

These findings prove the potential clinical efficacy of ACT that involves TILs for these and other cancer types, even if the identification and expansion of specific antitumor TILs is the actual challenge for the next future of this treatment. ACT involving the entire repertoire of TILs is indeed limited by the time and the difficulties in isolating and expanding tumor-reactive functional cells [96]. Certainly, a prerequisite for ACT to be effective is isolating and thus obtaining, upon expansion, an adequate number of tumor-specific effector T-cells from the patient. The problem of isolating the entire population of TILs is that this population is composed by a heterogeneous group of different subpopulations. Memory T-cells, anergic T-cells, exhausted T-cells, and even a consistent amount of immunosuppressive Foxp3^+^ and Foxp3^−^ T-cells are present in the reservoir of total TILs in different cancer types, thus limiting the frequency of antitumor effector cells [97,98]. Furthermore, a long term ex vivo stimulation in order to expand the tumor-reactive subpopulation of TILs and thus getting a sufficient number of effector cells, drives cells toward exhaustion and a loss of the wide range of antigen-specific TCRs [81,97]. Notably, cases of recurrences in cancer patients due to antigen loss and neoantigen mutations were widely documented, supporting the limitation of using T-cell clones with a limited epitope recognition, as in the case of CAR-T cells. Moreover, tumor-associated antigens for different solid cancers are still not fully characterized [81,97].

Thus, the rational for an effective TILs-based ACT would be identifying and isolating the ideal T-cell subset capable of maximizing the tumor cells killing, while eliminating anergic and suppressive T-cells from the TILs repertoire. Therefore, it would be beneficial to look at possible biomarkers in order to identify tumor-specific T-cells without the necessity of identifying tumor-derived antigens epitopes. 

Of note, many efforts were made in order to recognize tumor-reactive T-cells. The identification of rare T-cells carrying neoantigen-specific TCRs relies on the detection of specific somatic mutations of cancer cells. The problem is that these mutations show a wide variability among different patients even when considering tumors with similar features. The process would indeed require advanced and expensive technologies that are not only labor intensive, but also time consuming. Thus, it would require additional weeks besides those covered by the routine TILs separation and expansion. Consequently, more efficient methods in terms of labor and time consumption for the identification and isolation of antitumor TILs are required to give this ACT approach a clinical feasibility and validity.

Following antigen-induced activation and stimulation, human effector T-cells undergo phenotypic and functional changes, including the upregulation of surface markers. As previously discussed, the overexpression of CD137 receptor on antigen-specific activated T-cells provides the opportunity to clearly identify and isolate tumor-reactive effector cells by magnetic separation or fluorescence-activated cell sorting (FACS), thus removing the need of an ex vivo restimulation with defined antigen in order to reveal this subset of T-cells. A short term coculture with tumor cells in this case would be needed just in order to further stimulate the upregulation of CD137, since it was previously described that this receptor is strongly upregulated upon T-cells culturing together with tumor cells that express shared and neoantigenic peptides [64,66]. In addition, CD137^+^ isolation would not require samples rich of TILs, which may be difficult to obtain from surgical resections, in order to be sure to get the adequate number of effector cells. In fact, their isolation, that is possible not only at the tumor site but also from lymph nodes and even from peripheral blood, would be a precise isolation of tumor-reactive cells that need just to be expanded.

Based on these observations, Seliktar-Ofir et al. have reported for the first time a simple, robust, and fast method that allows the tumor-reactive enrichment for a therapeutic purpose [99]. They demonstrated that only 30.1 ± 25.9% of the TILs expressed CD137 after cocultures, but almost all cells expressed the receptor after the selection. In addition, they showed that CD137^+^ selected T-cells had an increased antitumor reactivity and contained a higher frequency of both shared and neoantigen reactive T-cells [99]. In this way, the CD137^+^ selection enables the isolation of antitumor effector T-cells without the need of knowing the epitopes of tumor-derived antigenic peptides.

These promising findings open novel prospective for the optimization of ACT in the near future.

## 5. CD137 in Current Clinical Practice: CD137-Targeting Antibodies and CD137 CAR-T Cells

In conclusion, it is important to mention the CD137-based approaches that are currently available for the clinical practice. Nowadays we have two different strategies involving the CD137 receptor [44]. First, the CD137 domain was included in the chimeric antigen receptors (CAR) T-cells. Second, monoclonal antibodies (MoAb) as well as bispecific antibodies targeting the CD137 receptor were proposed in the clinic as a strategy of cancer immunotherapy in different cancer types. 

The only approach involving the CD137 receptor that has so far been approved by the FDA is the CAR-T cells therapy targeting the CD19 for the treatment of pediatric B-cell leukemia and refractory B-cell lymphoma [100]. In this approach, the introduction of the CD137 intracellular domain, that is able to keep the survival signaling in these engineered T-cells, led to an increased persistence of therapeutical CAR-T cells when compared to those including the CD28 domain [101]. Additionally, CAR-T cells containing the CD137 signaling domain showed an advantage for both a metabolic point of view and also an advantageous epigenetic switch that increased a memory development by CAR-T cells [102].

Regarding the use of anti-CD137 MoAb in a clinical setting has raised particular interest, since it appeared capable of having a beneficial effect on tumor regression and oncological patient response, both when used as single agent and even more when combined to other drugs or anticancer treatments [44,54,55,65,103,104,105,106,107,108,109,110,111,112,113,114,115,116,117,118,119,120,121,122,123,124,125,126,127,128,129,130,131,132]. This clinically relevant effect is due to the ability of the antibody of stimulating an anticancer response that relies on effector T-cells [44,111]. After the stimulation with this MoAb, are not only the antitumor effector T-cells to be expanded, but importantly increased is also the frequency and the activity of tumor-specific memory T-cells that are able to confer a prolonged protection against cancer cells and thus a prolonged antitumor response [54]. 

Interestingly, the use of the anti-CD137 MoAb turned capable of reducing the activation and proliferation of both regulatory T-cells and myeloid-derived suppressor cells [55,103,110,133,134], that recently emerged as not only able to impair T-cell function but also of compromising the ICI efficacy through the impairment of DC function in cancer [135]. In addition, it was shown that the treatment with an anti-CD137 is able to increase the recruitment of effector T-cells inside the TME by targeting the receptor expressed by tumor-associated endothelial cells [104].

The mechanisms that can suppress antitumor immune responses in the context of cancer are various and this gives the rationale to the use of CD137 targeting in combination with other strategies aimed at removing these regulatory constraints [44,109,111,112,113,117,136,137].

So far, two different CD137-targeting agonistic MoAb have been introduced in the clinic: utolimumab (PF-05082566), a humanized IgG1 MoAb and urelumab (BMS-663513), a fully human IgG4 MoAb [44]. Being agonistic antibodies, both utolimumab and urelumab activate the CD137 signaling, promoting T-cell survival, proliferation, and cytokine production; while utolimumab blocks the CD137-CD137L interaction, urelumab, that appears to be a weaker agonist, does not prevent this engagement [44]. Phase I and II trials that tested the efficacy of urelumab in oncologic patients with advanced cancer demonstrated a notable effect of the MoAb as single agent, even if the trials were suspended due to the adverse liver inflammation that was noticed in treated patients [56]. The same adverse effects on liver inflammation induced by urelumab at doses of ≥1 mg/kg were spotted in a safety analysis that showed how a safe usage of the drug needed a marked decrease of the dose to 0.1 mg/kg [138]. Unfortunately, this dose decrease reduced the efficacy of urelumab as single agent, but permitted its use in combination with the anti-PD-1 targeting that gave promising clinical results in patients with advanced melanoma [57]. The ongoing trial that are testing Urelumab in combination with other agents are reported by Etxeberria et al. in Table 1 of their review [44]. 

Differently from urelumab, utolimumab showed no dose-related toxicity [44]. As single agent, it showed just little benefit in terms of clinical outcome for advanced cancer patients in phase I trials and this should probably be referred to its weakness as CD137 agonist [44,139,140]. On the contrary, Phase I trials testing utolimumab in combination with pembrolizumab [141,142] or with mogamulizumab in advanced solid tumor patients [143] and with rituximab in patients affected by relapsed or refractory non-Hodgkin’s lymphoma [144] showed no toxicity and thus safety and tolerability. In addition, promising antitumor effects and clinical responses were shown in all these studies. Other combination strategies that are currently ongoing in different trials are reported by Etxeberria et al. in Table 1 of their review [44].

Regarding bispecific antibodies, various approaches were recently hypothesized and are now under evaluation. In particular, the first and most advanced on those introduced the possibility of targeting the CD137 receptor on one side, and a tumor antigen (HER2) on the other side, in order to target this activation receptor specifically in the context of the TME [145]. This treatment has already shown clinical benefits in a Phase I trial in patients with HER2^+^ malignancies [146]. In line with that, other bispecific antibodies targeting the CD137 receptor together with other tumor antigens or the cancer associated fibroblast marker FAP (fibroblast activation protein), were tested in preclinical settings, showing promising results [147,148,149]. Since activated CD137^+^ T-cells can still be ineffective due to the exhaustion induced by the tumor, bispecific antibodies targeting the CD137 receptor together with PD-L1 are actually under evaluation (NCT03809624).

The weakness of using CD137-targeting in vivo should refer to the fact that the effectiveness of this targeting largely relies on the baseline presence of a sufficient number of naturally occurring antitumor CD137^+^ T-cells that are available to be unleashed and that are not suppressed by other mechanisms. Unfortunately, as previously reported, this scenario does not often reflect what really happens in cancer patients and thus requires combination of this approach with other strategies. 

## 6. Conclusions

With these emerging findings it is becoming clear that CD137^+^ T-cells play a pivotal role in exerting a specific antitumor activity as the final effectors of the complex immune response. Since most of the drugs that are available today for the treatment of oncological patients have the immune system as their target or, at least, rely on the patient’s immune system fitness in order to eliminate cancer cells, this population starts raising particular interest as a possible predictor of the responses to anticancer treatments. Moreover, their ability to predict the outcome and to monitor the responses to immune-based drugs, could be useful to aid clinicians in better stratifying patients and consequently selecting the best therapeutic approach. However, more efforts will be required in the future to validate their role as a possible biomarker in various cancer types

In addition, it will also be critical for the success of immunotherapies to optimize novel strategies of immunotherapy that have CD137^+^ T-cells as their protagonist. In fact, even if protocols for their isolation, expansion, and targeting have already been standardized and proven to be effective, other trials involving CD137^+^ T-cells based ACT and anti-CD137 agonistic moAb, will be required in the near future to probe their potential in different clinical settings. 

In conclusion, in light of the evidence reported in this review, it is important to state that trying to directly expand in vivo the antitumor effector T-cell population thanks to single agent strategies is not efficient in terms of patients’ clinical outcome. This is probably due to the fact that these cells need to reach the tumor area and then proliferate and exert their effector functions in order to observe a clinical response. An altered vascularization, T-cell exhaustion, and an immune suppressive tumor microenvironment, that are able to impair immune responses, are just some of the reasons why single agent strategies have recently proven to be ineffective [135,150,151,152,153,154]. In this scenario, the possibility of uncovering a receptor that characterizes the real antitumor T-cells population, will allow to target more precisely these cells with a combination of different strategies aimed to facilitate cancer cells’ elimination and preventing immune suppression. 

Of note, it is important to highlight that other second line costimulatory receptors, as in the case of OX40 (CD134, TNFRSF4), showed to be regulated in a similar way if compared to CD137 receptor, having thus a similar therapeutic potential [155,156,157,158] and questioning the uniqueness of the CD137 receptor in pointing out the natural occurring antitumor T-cells. However, the aim of this review was to present the latest findings on how the CD137 receptor is alone able to identify a population capable of recognizing and eliminating cancer cells, thus discussing the potential use of this population as a biomarker to evaluate the patients’ capability to respond to immune-based therapies as well as the potential benefit of introducing CD137^+^ T-cells as protagonists of novel strategies for the treatment of oncologic patients.

## Figures and Tables

**Figure 1 cancers-13-00456-f001:**
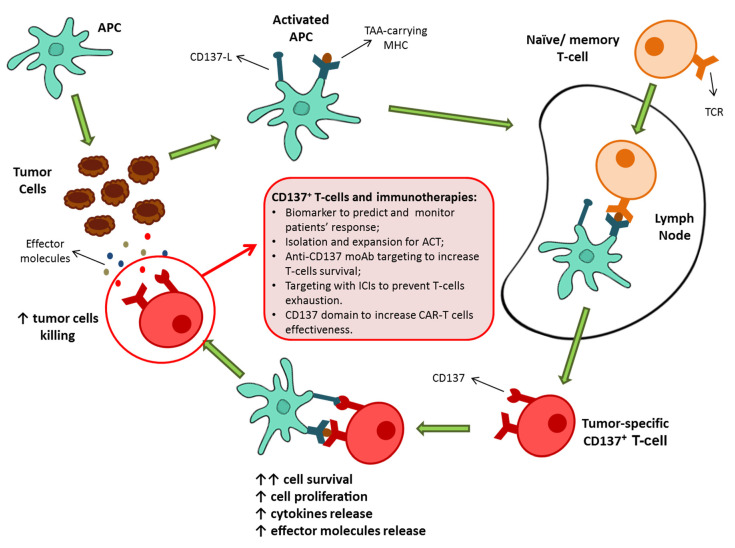
Schematic representation of CD137^+^ T-cell activation as a result of priming by TAA-carrying activated APC. The consequence of the CD137/CD137-L engagement is a marked increase in cell survival, followed by an increased proliferation, cytokine production, and effector molecules release. Then, possible roles of CD137^+^ T-cells population in the present and future of immunotherapy. APC, antigen presenting cell; TAA, tumor-associated antigens; MHC, major histocompatibility complex; ACT, adoptive cell therapy; moAb, monoclonal antibodies; ICIs, immune checkpoint inhibitors; CAR, chimeric antigen receptor.4. CD137^+^ T-cells can predict cancer patients’ response to immune-based therapies.

**Table 1 cancers-13-00456-t001:** Summary of the results showing the power of CD137^+^ T-cells population as a biomarker able to predict and monitor patients’ response to different immune-based therapies in various tumor models.

Cancer Type	Treatment	Results	References
**Metastatic NSCLC**	Anti-PD-1 ICIs	Higher percentages of CD137^+^ T-cells in PBMC predicted a prolonged patients’ OS and PFS.	[67,68] Ugolini et al., 2020
**Metastatic RCCC**	Anti-VEGF-R TKIs and anti-PD-1 ICIS	Percentage of CD137^+^ T-cells in PBMC decreased during patients’ progression.	[69] Zizzari et al., 2018
**Metastatic RCCC**	TKIs	Higher percentages of CD137^+^ T-cells in PBMC were associated with responder patients.	[70] Zizzari et al., 2020
**Metastatic Melanoma**	Anti-PD-1 ICIS	CD137 mRNA levels at the tumor site were positively associated with a prolonged OS, PFS, and a better response to the therapy.	[71] Fröhlich et al., 2020

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
