# Peer review of "CD137+ T-Cells: Protagonists of the Immunotherapy Revolution"

_cancers, 2021, doi:10.3390/cancers13030456_

Round 1

Reviewer 1 Report

The authors have provided a concise perspective about the role of the CD137 molecule as an effective proliferative and anti tumor molecule , involved in the effector function of T cells. It is important to note that the authors have provided insight into the role of CD137 in the expansion and as noted in CAR-T therapies for anti-cancer treatment. The only suggestion is for the authors to look into language correction at various places within the manuscript.

Reviewer 2 Report

This description of CD137 in cancer immunity is one of many similar review articles. There are some much more comprehensive presentations of the role of CD137 in immunotherapy. However, each review article is valuable and may add something new.

Thus, in general, this paper is worthy to publish but needs some corrections.

  1. Abstract should be more informative; I suggest adding one sentence summarizing each paragraph
  2. It will be valuable to present results of the studies performed in different tumors in the table with authors names and number of references
  3. It should be pointed that the CD137 antibody is an agonistic antibody and explain the mechanism of agonistic antibody action
  4. The figure needs description, not only title. The more- please describe the second link apart from CD137-CD137L: receptor and ligand on APC and T cell.
  5. In the chapter 5, from the line 156 the Authors repeat many times “In fact”. Is it intentional? If not, it will be better to correct the style.

Reviewer 3 Report

Ugolini and Nuti discuss the role of CD137+ T cells for different T cell based concepts for the immunotherapy of cancer. They present a lot of evidence from literature describing the prognostic value of CD137+ T cells for cancer patients. The authors propose CD137 to be a marker of tumor specific T cells. CD137, however, appears to be a late co-stimulatory receptor of previously activated T cells.  CD134 (OX40) that is also a second line co-stimulatory receptor of activated T cells appears to be regulated in a similar fashion. Accordingly, there is growing evidence in the literature that agonistic OX40 antibodies can act in a similar fashion as anti-CD137 antibodies. In light of these results and redundant co-stimulatory pathways the unique properties of CD137+ T cells are questionable and should be discussed.

Major

To improve readability of the manuscript I highly recommend proofreading by a native speaker.

Minor

Page 2

 line 66: …”with anti-CD137 monoclonal antibodies…” should read ….”with an agonistic anti-CD137 monoclonal antibodies….”

Page 4 ,

second paragraph (lines 120-135) should be re-written.

Page 6,

line 221 “They proved that….” Should be read they demonstrated that….”

Page 7

line 292 “In fact, the presence of this population begins to a good candidate in predicting patients…”  should be read “In fact, the presence of this population begins to be a good candidate in predicting patients…”

The conclusion section should be rewritten.

Reviewer 4 Report

Alessio Ugolini and Marianna Nuti reviewed the current approaches in immunotherapy specifically by the use of CD137+ T-cells as protagonists of immunotherapy. This review is important as it brings out the point that CD137+ T cells as the main effector population activated against cancer cells. The authors have also discussed the possibility of using CD137 as a biomarker to evaluate the fitness of the patients’ immune system in order to predict the outcome of immune-based treatment against cancer. This is aimed at ensuring that the applied immunotherapies used with or without other immune-checkpoint inhibitors will be effective and beneficial to the patient. Additionally, the authors also talk about the use of anti-CD137 MoAb that can be used as anti-tumor agents as well as its beneficial use in intracellular domain of CAR T cells.

Major comments are as follows:

  • English proof-reading including grammar, prepositions and consistency of used abbreviations is highly recommended. Of note, could be useful to use an English native speaker for proof-reading of document.
  • Recommendation in general:
  • Introduction of abbreviations throughout the text for the reader to be able to follow easily
  • Please check formatting errors throughout the text
  • g Italicise in vivo, ex vivo
  • Single spacing between words
  • CD137+, the plus sign should be used consistently either as superscript or normal sign
  • Consistency in grammatical tenses if it is past tense, e.g. Page 1 line 70-71 “….was made when CD137+ T-cells have been clearly identified as those cells that were not only….”

  • In general, since the review focuses on the use of CD137+ T cells and or CD137 receptor in immunotherapy, kindly provide abit more information on the signalling of CD137 with its ligand or the effects of the agonistic antibody in expansion of T cells, survival, and its cytolytic function in order for the reader to be able to fully understand the beneficial effects.
  • Page 1 line 20-25, the sentence is too long. Please break it down
  • Page 1 line 20-25, kindly explain what you mean by “unleashed from a regulatory constrain” could you mention any or some of the regulatory constrain
  • Page 1 line 27, proper fold/sequence for the reference [1,2,11,3–10]
  • Page 1 line 40: CD137, expression on CD8+ T-cells is earlier rather than “faster”
  • Page 2 line 47-51, the sentence is too long. Please break it down in order to provide clarity of the message being conveyed
  • Page 2 between line 60 and 61, kindly include a bridging sentence. The subsequent sentence is not related to the former?
  • Page 2 line 60- 69, this sentence could be linked to the paragraph starting with line 46 since the effects in both CD8+ and CD4+ T-cell have also been described in the references (31-34)
  • Page 2 line 71-72, the sentence “…..was made when CD137+ T-cells have been clearly identified as those cells that were not only terminally differentiated effector cells but, above all, were also antigen-specific T-cells…” do you intend to mean that CD137+ T-cells are terminally differentiated effector cells and yet are still antigen-specific T-cells?
  • The general flow of the text in the first part (line 46-74) could be improved to have a better structure in order to be more elaborate. Suggestions of the flow according to the already pointed out information on your text could follow this format:
  1. Expression in CD4/CD8+ T cells
  2. Effect in antiviral T-cell responses
  3. Effect in tumour T-cell responses
  4. Utilisation as a strategy in depletion of activated alloreactive T cells during allogeneic hematopoietic stem cell transplantation
  • Page 2 line 76-81, please break this sentence. Additionally, grammatical check needed- “The discover that CD137 is expressed by most of activated and antigen-specific…” and “……even if they were present initially at a very low frequency, although this required an ….”
  • Page 2 line 78, MHC-restriction elements; perhaps complex could be better?
  • Page 2, Preferentially use either antigen-expressing cancer cells or cells expressing tumor antigen, usage of both terms can be a bit confusing to the reader
  • Page 2 line 90 “.. ..need of knowing.. - better to use the word knowledge?
  • Page 3 line 101”…. producing big amount” - use better term for big?
  • Page 3 line 103, indicate which subset of “human T cells” were transferred to the immunodeficient mice to have a clear comparison of the benefits of CD137+ T cells in inhibiting tumor growth
  • Page 3 line 105, “….reactivity; in fact, being CD137 expression a rapid and transient event upon specific activation…”please clarify what you mean or re-write to bring out the point clearly
  • Page 3 line 105, “tumor cells that do not expressed a determined antigen or that mutated their antigens” please explain what you mean. Did you intend to mean tumor cells without a specific antigen or those that express a mutated antigen?

Figure 1:

  • Kindly include a legend to Figure 1 in order to briefly explain the processes and label the figure more precisely
  • Abbreviation ICI (immune checkpoint inhibitors) was not introduced
  • There is a double arrow on increase in cell survival
  • Please provide a brief description of what you mean or how CD137+ T cells are applied as biomarkers to predict and monitor patients’ response. Do you intend to mean it is a biomarker in terms of responses to tumor eradication or in enhancing effectiveness of immune checkpoint inhibitors or a biomarker for tumor-reactive cells?
  • In your previous study, you clearly described “autoantibody IgM-FR as possible biomarker of T-cells fitness and their ability to respond to ICIs in NSCLC and that IgM-RF appears to be a predictive factor of early progression”.

Is it possible that only CD137+ T cells frequency parameter can be used as a stand-alone biomarker to effectively predict and monitor patient’s responses as mentioned in this context in Figure one? Please elaborate this point

  • Page 4 line 117, the use of: “despite these clear results” can be used later in the text after discussion of the results from studies that have reported the portrayed phenomenon, so that the reader does not get the conclusion before the context.
  • Page 4 line 123-126, I believe you mean that patients with autoantibodies IgM-RF have a reduced frequency of tumour-reactive CD137+T-cells?. However, I do not understand what you mean by as a “consequence of impaired the migration of naïve and central memory T-cells” is the migration impaired to the peripheral blood as you claim or to sites of the tumor?

In connection to this, you described in page 3 line 95-100 that Ye et al. compared frequency of CD137+ T-cells in different locations and was reported to be decreased toward the periphery, does this rule out the claim that CD137+ T-cells are reduced in peripheral blood as a pathophysiology due to cancer rather than due to autoantibodies IgM-RF? Or most cancer patients are positive for IgM-RF autoantibodies?

  • Page 4 lines 132-133, kindly use more scientific term to replace or be more precise “these kinds of treatments” as well as the term “useless”. Re-write the sentence if possible.
  • Page 4 lines 136-138, the message you would like to convey does not come out clearly in the sentence. Please re-write the sentence
  • Page 4 lines 144, Could be that the word improves or enhances is missing in this sentence “indirect way, improves the immune system ability of killing tumor cells”
  • Page 4 lines 150, Was the effect of TNFRSF9 low methylation levels and the subsequent increased mRNA expression determined in isolated CD137+ T cells? This was not clear. If not, how was this concluded that CD137+ T-cells improved survival for cancer patients (line 148-150)?
  • Page 4 lines 154 activated T effector cells preferably activated effector T cells?
  • Page 4 and page 5 under section “ CD137+ T-cells-based therapies as possible protagonists for future immunotherapies” Interesting and elaborate information on TIL. However, the focus is largely on TILs. Perharps you could summarise or highlight the main or important information on TIL as it is not the main focus since you already introduced TIL in section 3. Possibly you could expound more on correlation of CD137 expression levels on tumour-infiltrating T cells and the outcome/ success rates in anti-tumor reactivity as you have done in lines 220-226 (only a suggestion).
  • Alternatively, you can have this title: CD137+ T-cells-based therapies as possible protagonists for future immunotherapies as the main sub-heading and then discuss TIL, CAR-T cells and possibly (optional) mention additional T-cell based therapies/therapy e.g. Natural killer cells which also express CD137, as subsequent minor sub-headings.
  • Page 4 lines 160 “In particular, they raise a particular interest…”. Repetitive use of particular in the sentence
  • Page 6 line 229 - 234 In the first paragraph under, CD137 targeting and CD137 CAR-T cells, in addition to the approaches encompassing monoclonal antibodies (MoAb) targeting the CD137 receptor, you could also mention briefly bispecific antibodies (optional)
  • Page 6 line 232 and 233, the use of “On one side” in consecutive sentences is repetitive
  • Page 6 line 233 and 234, kindly mention briefly the tumor models or types of cancer that are targeted by anti-CD137 MoAb
  • Page 6 line 238, could not clearly understand what you meant by CD137 intracellular domain “…able to exert the survival signal…” is this in the context of CD137 (4-1BB) signaling improves T cell persistence compared to CD28 signaling?
  • Page 6 line 242-243 “beneficial effect on tumor growth and oncological patient response” this sounds contrary to what you might mean in terms of; anti-CD137 MoAb should not support the growth of the tumor rather the opposite
  • Page 6 line 246 ability of the antibody of stimulating an anti-cancer response - replace “of” with “in”
  • Page 6 line 260, the mentioned CD137-targeting MoAbs have different mechanisms of action, could you briefly mention this?
  • Page 6 line 262-264, It would be great if you could provide information on what amount of MoAb urelumab resulted to adverse effects of liver inflammation and to what reduced amount resolved this adverse effect
  • Page 6 line 273-277, the sentence is quite long. Please break it down
  • Page 6 in the conclusion section, please revisit grammatical tenses

Overall, do you consider CD137+ T cells as a tumor infiltrating immune cell subset? Please comment on that and if it circumvents the problem of T-cells or immune cells not being able to reach the tumor site because of the immunosuppressive tumor microenvironment.

Reference list – According to the guidelines of the journal, please use consistent formatting throughout. “It is essential to include author(s) name(s), journal or book title, article or chapter title (where required), year of publication, volume and issue (where appropriate) and pagination. DOI numbers (Digital Object Identifier) are not mandatory but highly encouraged”. 

Some references include the volume/the issue number such as “ Cancers (Basel). 2020, 12, 2620” but the rest do not have this feature. Others are incomplete eg 4. Science (80-. ). 2006, Most have DOI but others do not

Round 2

Reviewer 2 Report

All corrections are adequate